# Systematic Literature Review on Methods of Assessing Carrying Capacity in Recreation and Tourism Destinations

**Zamru Ajuhari** [1,*], **Azlizam Aziz** [1], **Sam Shor Nahar Yaakob** [1], **Shamsul Abu Bakar** [2,*] and **Manohar Mariapan** [1]

1   Department of Recreation and Ecotourism, Faculty of Forestry and Environment, Universiti Putra Malaysia, Serdang 43400, Selangor, Malaysia
2   Department of Landscape Architecture, Faculty of Design and Architecture, Universiti Putra Malaysia, Serdang 43400, Selangor, Malaysia
*   Correspondence: zamruajuhari@gmail.com (Z.A.); shamsul_ab@upm.edu.my (S.A.B.)

**Abstract:** Carrying capacity is paramount to recreation and tourism management, which depends on sustainability between resource protection and experience quality. Many studies have examined carrying capacity from several perspectives, but the various methods of assessing carrying capacity have not yet been reviewed. The purpose of this study is to assess the methods of carrying capacity, their trend, and the assessment of carrying capacity made by each method. From the three scientific repositories used in this research, 100 original research papers were included in the review process. A total of 24 methods were recorded. The normative approach and Cifuentes Arias' method were found to be the two main methods of determining carrying capacity. From the assessment of carrying capacity and the origin of each method, two fundamentals of carrying capacity emerged, and their differences and limitations are discussed. In addition, the study found that the carrying capacity employed in tourism destinations was formulated by complex variables that may require political interventions to ensure their success. Most of the research reviewed here focuses on the social aspects of carrying capacity, thus leaving room for future research. This study should benefit academics, policymakers, and resource managers by comprehensively analyzing the methods, limitations, and directions of future research in carrying capacity studies.

**Keywords:** carrying capacity; sustainability; recreation; tourism; normative approach; Cifuentes Arias' method; systematic literature review

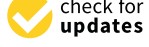



## 1. Introduction

Recreation and tourism resources are usually natural settings that are preserved to safeguard ecological, cultural, aesthetic, and scientific values. Crucially, such resources are a platform for promoting environmental values. Participation in recreational and tourism activities invokes a sense of attachment to a place that encourages appreciation toward nature and nurtures pro-environmental actions [1–5].

Recreation and tourism are now parts of the modern lifestyle. As the population grows, the demand for recreation and tourism has become a challenge to recreation and tourism management, as recreation and tourism resources are consistently challenged by visitors' depreciative behaviors [6–13] and crowding [14–16].

Impacts resulting from heavy visitation have caused undesirable changes to vegetation and plant diversity [17,18], erosion of the soil [19,20], diminished water quality [21], and disturbance to wildlife [22–25]. At the same time, recreation negatively impacts visitors' recreational enjoyment [26–28] as well as their loyalty and intention to revisit a place [29,30]. To address the increasing impacts of recreation activities, carrying capacity was introduced in recreation resource management in the early 1960s. Some earlier contributors (e.g., [31–48]) have paved the way for many carrying capacity studies in recreation and tourism resources today.

Previous works that have systematically reviewed carrying capacity have provided valuable insights into the hotspots in carrying capacity research topics [49], such as the use of a digitalization approach in measuring carrying capacity [50], carrying capacity from the regional perspective [51,52], and the concept and model of tourism carrying capacity [53]. Butler's review [54] underlined the future direction of carrying capacity, and Wei et al. [55] emphasized the specific types of carrying capacity. Nevertheless, even though previous systematic reviews have significantly contributed to the advancement of carrying capacity studies, it is worth noting that the variety of methods used in carrying capacity determination and their fundamentals have yet to be comprehensively documented and academically discussed in one publication.

Given these gaps in the literature and the implementation of visitor capacity as essential in recreation and tourism resources, this study systematically reviews original research papers under the domain of carrying capacity studies to assess the following: (1) the methods that have been used to assess carrying capacity, (2) the trend of each method, and (3) the assessment of carrying capacity that was made by each method. Based on the results, this study outlines the methods' differences, fundamentals, limitations, issues, and gaps for future research. This study provides an up-to-date overview of the existing literature on carrying capacity studies from the past two decades and the gaps in the current knowledge that warrant more future studies.

## 2. Materials and Methods

This systematic literature review was conducted following the Preferred Reporting Items for Systematic Review Recommendations (PRISMA) guidelines [49,50,56,57] and adapted for this study. Research papers were obtained by searching three repositories of scientific publications: Google Scholar, SCOPUS, and MDPI. The keywords used in the searches were "recreation and tourism carrying capacity" OR "recreation and tourism visitor capacity" OR "recreation and tourism social carrying capacity" OR "recreation and tourism physical carrying capacity" OR "recreation and tourism ecological carrying capacity" OR "ecotourism carrying capacity" OR "recreation and tourism environmental carrying capacity" OR "recreation and tourism carrying capacity standard" OR "recreation and tourism carrying capacity threshold" and "recreation and tourism visitor use management".

The criteria for selecting the research papers were that they had to be published from 2000 to October 2022 and written in English. This period was chosen because the interest of this study was to assess the methods of determining carrying capacity in the past two decades of research. In addition, it was also consistent with the recent systematic literature review papers that reviewed carrying capacity studies between the past 20 and 26 years [49,50]. The searches were carried out progressively between January and October 2022. For each repository, references that did not meet the preliminary criteria were excluded. The search process was as follows.

1. Google Scholar: A search was carried out for each keyword, and the results were screened by the custom range of years of publication, namely, 2000 to 2004, 2005 to 2009, 2010 to 2014, 2015 to 2019, and 2020 to 2022. The search criteria were sorted by "relevance", "any type", and "include citation". The list of papers for each search was recorded and downloaded until page 10.
2. SCOPUS: A search was carried out within "Article title, Abstract, Keywords". Results were screened based on the journal in which they were published, open access and open archive status, the year of publication, the research article itself, and whether they were published in English. The results were downloaded and listed in a dataset.
3. MDPI: A search was carried out for journal type. The results were screened based on the article, year of publication, and whether they were in the English language. The results were downloaded and listed in a dataset.

Research papers were downloaded once per database, even if they appeared across the three repositories. The search focus was on the content of the research papers; therefore,

research papers with titles and keywords that did not match the search were included in the pool. A total of 3995 research papers were initially recorded (Figure 1).

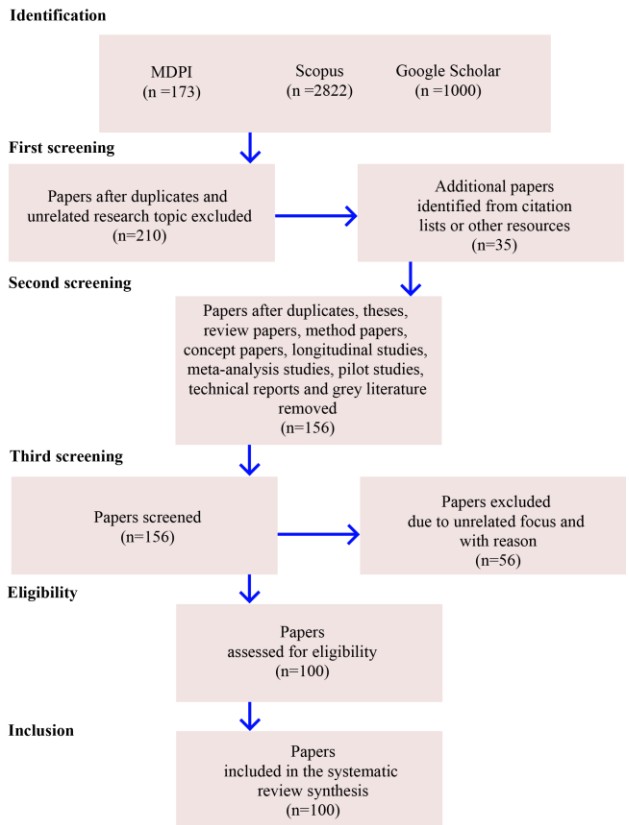

**Figure 1.** Modified PRISMA flowchart outlining the process and actions taken to compile research papers. Note: n refers to the number of research papers.

The second screening process was conducted based on an approach by Ballantyne et al. [58]. In this case, the research papers had to be original, peer-reviewed, and related to the research topic. Theses, review papers, method papers, concept papers, longitudinal studies, meta-analysis studies, pilot studies, technical reports, and gray literature were excluded; however, they were used to find other academic-related journals. The second screening process was carried out by the five authors of this study separately. As a result, a total of 245 publications were carried onto the third screening process.

Lastly, the five authors screened the research papers together to reach a consensus regarding the final number of papers to be included in the review. During this process, research papers related to the formulation or the determination of indicators for carrying capacity and carrying capacity monitoring were excluded, as this was not the main interest of this study. As a result, 100 research papers were eventually identified and included in the systematic quantitative literature review. These research papers were sorted according to the year of publication, first author, journal name, publisher, country, area of study, type of settings, carrying capacity assessments, and method used to determine the carrying capacity.

## 3. Results

The details of the publication, carrying capacity assessment, and methods for carrying capacity determination are detailed in this section. Supplementary Materials list the 100 included sources and 145 excluded sources for the systematic literature review.

### 3.1. Methods of Carrying Capacity Determination

Of the 100 research papers reviewed, 24 methods of assessing carrying capacity were recorded. The methods were then divided into four categories according to their similarity and origin (Table 1). The first category was the normative approach, which specifically employs visual aids and a social norm curve for carrying capacity determination. The second category originates from Cifuentes Arias' method, and the third category uses a method that combines the two previously mentioned methods. The last category was grouped as "other method", consisting of the individual method that assessed carrying capacity from economic perspectives, formulas, models, frameworks, environmental resources, and visitor surveys. Of the 24 methods recorded, the normative approach (f = 45) and Cifuentes Arias' method (f = 30) were the two most prominent methods used in carrying capacity determination.

**Table 1.** The methods for carrying capacity determination.

| Method | Frequency |
|---|---|
| **Normative approach** | |
| Normative approach (visual approach and social norm curve) | 29 |
| Normative approach and regression model | 3 |
| Audio recording and social norm curve | 3 |
| Normative approach and willingness to pay | 2 |
| Visual approach and stated choice analysis | 2 |
| Normative approach and contingent valuation method | 1 |
| Visual approach and Bayesian generalized linear mixed modeling | 1 |
| Normative approach and level of service of shared-use paths | 1 |
| Visual approach and regression model | 1 |
| Scenario of encounter and social norm curve | 1 |
| Normative approach and audio recording | 1 |
| **Cifuentes Arias' method and its variants** | |
| Physical carrying capacity (PCC), real carrying capacity (RCC), and effective carrying capacity (ECC) formula | 21 |
| PCC and BCC (beach carrying capacity) formula | 2 |
| Carrying capacity formula akin to the PCC formula | 2 |
| Formula akin to PCC and RCC | 1 |
| Videometric and BCC formula | 1 |
| PCC and RCC formula | 1 |
| Ecotourism carrying capacity formula akin to the formula of PCC | 1 |
| RCC formula | 1 |
| **Cifuentes Arias' method + normative approach** | |
| PCC, RCC, ECC, and visual approach | 1 |
| PCC and normative approach | 1 |
| **Other methods** | |
| Choice experiment method (willingness to pay) | 2 |
| Proportion curve of divers that damaged hard coral vs. number of divers in the study area and regression model | 2 |
| Tourism economy carrying capacity system formula, resource carrying capacity system formula, and tourism carrying capacity formula | 1 |
| Fuzzy coefficient linear programming model | 1 |
| The linear programming technique | 1 |
| Pressure-state-response framework—tourism carrying capacity index | 1 |
| Time-for-time substitution model and individual-based, system dynamics model | 1 |
| Carrying capacity value stretch model | 1 |
| Stated choice modeling | 1 |
| Spatial model and optimum boating density formula | 1 |
| Ecological carrying capacity formula and social carrying capacity formula | 1 |
| Tourism environmental carrying capacity formula | 1 |
| Crowding perception preferred, acceptable, and intolerable threshold values' estimations and Wilcoxon test | 1 |
| Residents' quality of life, visitors' quality of experience, and regression model | 1 |
| Visitor survey, satisfaction, and discriminant analysis | 1 |
| Crowding perception | 1 |
| Daily visitation model, perceived level of satisfaction model, and regression model | 1 |
| Number of people acceptable (visitor survey) | 1 |
| Narrative/numerical research methods | 1 |
| Indifference curve analysis | 1 |
| Attitude of the locals, tourist–host inter-relationships, and effects of tourism on social behavior and values | 1 |

Note. n = 100.

### 3.2. Publication Trend of Each Method

Manning contributed five research papers using the normative approach method, and Needham and Hallo contributed three each (Table 2). Six authors contributed two papers each, and another 22 contributed one paper each since 2000. The publication year based on the normative approach method was spread across 22 years, with most papers published in 2012 and 2011 (11.11% each). Nearly 65% of the carrying capacity studies with this method were conducted in the USA, with over 65% being carried out in wilderness settings.

**Table 2.** Publication details of the normative approach method.

| Category | Frequency (%) |
|---|---|
| **First author** | |
| Manning, R.E. | 5 (11%) |
| Needham, M.D. and Hallo, J.C. | 3 (6.67%) |
| Six first authors each | 2 (4.44%) |
| 22 first authors each | 1 (2.22%) |
| **Publication year since 2000** | |
| 2011 and 2012 | 5 (11.11%) |
| 2005 and 2020 | 4 (8.89%) |
| 2009, 2010, 2013, 2015, 2016, and 2019 | 3 (6.67%) |
| 2002 and 2022 | 2 (4.44%) |
| 2000, 2004, 2006, 2018, and 2021 | 1 (2.22%) |
| **Country** | |
| USA | 29 (64.44%) |
| Canada | 4 (8.89%) |
| China | 3 (6.67%) |
| Mexico and South Korea | 2 (4.44%) |
| Sri Lanka, Japan, Malaysia, Italy, and Jordan | 1 (2.22%) |
| **Setting** | |
| Wilderness | 30 (66.67%) |
| Marine | 5 (11.11%) |
| Rural and urban | 2 (4.44%) |
| Beach, sinkholes, island/monument, coastal/monument, lake/reservoir, and beach | 1 (2.38%) |

Note. n = 45.

Apart from these studies, de Sousa contributed three research papers using Cifuentes Arias' method, and another 27 authors contributed one paper each using this method (Table 3). Next, 90% of the research papers using this method were published post-2011, mostly in 2021 (23.33%), 2019 (16.67%), and 2022 (13.33%). Geographically, 30% of the carrying capacity studies using this method were conducted in Brazil, China, and Iran, with over 60% being carried out in beach and urban settings.

**Table 3.** Publication details of Cifuentes Arias' method.

| Category | Frequency (%) |
|---|---|
| **First author** | |
| de Sousa, R.C. | 3 (10%) |
| 27 authors each | 1 (3.33%) |
| **Publication year since 2000** | |
| 2021 | 7 (23.23%) |
| 2019 | 5 (16.67%) |
| 2022 | 4 (13.33%) |
| 2015 | 3 (10%) |
| 2008, 2013, 2017, and 2018 | 2 (6.67%) |
| 2011, 2014, and 2020 | 1 (3.33%) |

**Table 3.** *Cont.*

| Category | Frequency (%) |
| --- | --- |
| **Country** | |
| Brazil and China | 5 (16.67%) |
| Iran | 4 (13.33%) |
| Mexico, Spain, and Italy | 3 (10%) |
| Indonesia | 2 (6.67%) |
| Portugal, Argentina, Serbia, Jordan, and India | 1 (3.33%) |
| **Setting** | |
| Beach | 13 (43.33%) |
| Urban | 6 (20%) |
| Wilderness | 5 (16.67%) |
| Marine and rural | 2 (6.67%) |
| Desert/monument and island/coastal | 1 (3.33%) |

Note. n = 30.

The studies that utilized a combination of Cifuentes Arias' method and the normative approach were published by one author each in 2011 and 2013, respectively, and conducted in Portugal in a beach setting (Table 4). In addition, 23 studies employed another 21 methods of assessing carrying capacity. Most of these studies were conducted in China (21.74%) and the USA (13.04%), in wilderness (34.78%) and urban (21.74%) settings.

**Table 4.** Publication details of Cifuentes Arias' method + normative approach and other carrying capacity methods.

| Category | Frequency (%) |
| --- | --- |
| **Cifuentes Arias' method + normative approach \*** | |
| **First author** | |
| Zacarias, D.A. | 1 (50%) |
| Silva, S.F. | 1 (50%) |
| **Publication year since 2000** | |
| 2011 and 2013 | 1 (50%) |
| **Country** | |
| Portugal | 2 (100%) |
| **Setting** | |
| Beach | 2 (100%) |
| **Other methods \*\*** | |
| **First author** | |
| Lawson, S.R. | 2 (8.7%) |
| 21 authors each | 1 (4.35%) |
| **Methods** | |
| Choice experiment method (willingness to pay) | 2 (8.7%) |
| Proportion curve of divers that damaged hard coral vs. number of divers in the study area and regression model | 2 (8.7%) |
| 19 methods each | 1 (4.35%) |
| **Publication year since 2000** | |
| 2020, 2012, and 2002 | 3 (13.04%) |
| 2017 | 2 (8.7%) |
| 2022, 2021, 2018, 2016, 2013, 2011, 2009, 2008, 2007, 2006, 2001, and 2000 | 1 (4.35%) |
| **Country** | |
| China | 5 (21.74%) |
| USA | 3 (13.04%) |
| Israel and Italy | 2 (8.7%) |
| 11 countries each | 1 (4.35%) |
| **Settings** | |
| Wilderness | 8 (34.78%) |
| Urban | 5 (21.74%) |
| Beach and marine | 3 (13.04%) |
| Rural and island/coastal | 2 (8.7%) |

Note: \* n = 2, \*\* n = 23.

### 3.3. Assessment of Carrying Capacity Made by Each Method

Sixty carrying capacity assessments were recorded for the normative approach method, representing recreation and tourism resources' social, resource, and managerial components (Table 5). Of these, 75% of the assessments evaluated social conditions, 20.59% assessed resource conditions, and 4.41% measured carrying capacity related to managerial conditions. The most frequent assessment centered on the number of people at one time (32.35%), the number of boats seen/number of boats at one time (8.82%), and the percentage of bare ground on a campsite (7.35%).

**Table 5.** Assessment of carrying capacity by normative approach method.

| Carrying Capacity Assessment | Frequency (%) |
|---|---|
| **Social** | |
| Number of people at one time | 22 (32.35%) |
| Number of boats seen/number of boats at one time | 6 (8.82%) |
| People per viewscape | 4 (5.88%) |
| Visitor-caused noise/human-caused noise | 3 (4.41%) |
| People/visitors/tourists per day | 2 (2.94%) |
| Number of people within a meter distance | 2 (2.94%) |
| Distance to watch wildlife | 1 (1.47%) |
| Size of boating groups | 1 (1.47%) |
| Minimum acceptable chance of catching selected types of fish | 1 (1.47%) |
| Minimum acceptable chance of seeing selected types of wildlife | 1 (1.47%) |
| Density of hikers | 1 (1.47%) |
| Percentage of time of human-caused noise | 1 (1.47%) |
| Number of cars per meter of distance on the road | 1 (1.47%) |
| Off-road vehicle density | 1 (1.47%) |
| Density of mountain bikers | 1 (1.47%) |
| Number of busses at wildlife stops | 1 (1.47%) |
| Number of kayaks per 1500 m | 1 (1.47%) |
| Number of vehicles per day | 1 (1.47%) |
| **Resource** | |
| Percentage of bare ground on the campsite | 5 (7.35%) |
| Amount of litter | 2 (2.94%) |
| Percentage of bare soil on the trail | 1 (1.47%) |
| Level of on-trail development | 1 (1.47%) |
| Level of off-trail management | 1 (1.47%) |
| Damage to vegetation and soil on the trail | 1 (1.47%) |
| Trail widening and deepening | 1 (1.47%) |
| Level of air quality | 1 (1.47%) |
| Amount of graffiti | 1 (1.47%) |
| **Managerial** | |
| Chances of obtaining permit for camping | 1 (1.47%) |
| Number of houses along the shore | 1 (1.47%) |
| Waiting time to receive permit | 1 (1.47%) |

Note. n = 68.

Concerning Cifuentes Arias' method, 30 carrying capacity assessments were recorded that focused on the social aspects of recreation and tourism resources (Table 6). Over 80% of the assessments centered on people/visitors/tourists per day, and 6.67% focused on the number of visitors in an area (m or ha). For Cifuentes Arias' method plus the normative approach, two assessments were recorded exclusively focusing on the people/visitors/tourists per day (100%).

**Table 6.** Assessment of carrying capacity by Cifuentes Arias' method and Cifuentes Arias' + normative approach *.

| Carrying Capacity Methods and Assessment | Frequency (%) |
| --- | --- |
| **Carrying capacity assessment of Cifuentes Arias' method *** | |
| People/visitors/tourists per day | 25 (83.33%) |
| Number of visitors in an area (m or ha) | 2 (6.67%) |
| People/visitors/tourists per year | 1 (3.33%) |
| Number of people at one time | 1 (3.33%) |
| People/visitors/tourists per month | 1 (3.33%) |
| **Carrying capacity assessment of Cifuentes Arias' method + normative approach **** | |
| People/visitors/tourists per day | 2 (100%) |

Note: * n = 30, ** n = 2.

Fifty carrying capacity assessments were recorded that used the other methods, representing recreation and tourism resources' social, resource, and managerial components (Table 7). Next, 50% of the assessments evaluated social conditions, 28% assessed resource conditions, 14% measured carrying capacity related to managerial conditions, and 8% considered carrying capacity as an index or value. The most frequent kind of assessment centered on people/visitors/tourists per day (18%) and the amount of litter (8%).

**Table 7.** Assessment of carrying capacity by other methods.

| Carrying Capacity Assessment | Frequency (%) |
| --- | --- |
| **Social** | |
| People/visitors/tourists per day | 9 (18%) |
| Number of visitors in an area (m or ha) | 2 (4%) |
| Number of people per meter | 2 (4%) |
| Number of other groups encountered per day while hiking | 1 (2%) |
| Likelihood of being able to camp out of sight and sound of other groups | 1 (2%) |
| Number of groups per day | 1 (2%) |
| Level of difficulty in obtaining a permit for an overnight wilderness trip | 1 (2%) |
| Number of guided dives per site per year | 1 (2%) |
| Local community level of tolerance | 1 (2%) |
| Number of boats seen/number of boats at one time | 1 (2%) |
| Number of vehicles per day | 1 (2%) |
| Intensity of restrictions regarding where wilderness visitors are allowed to camp | 1 (2%) |
| Optimum boating density in $km^2$ | 1 (2%) |
| Level of visitor awareness | 1 (2%) |
| Number of people at one time | 1 (2%) |
| **Resource** | |
| Amount of litter | 4 (8%) |
| Percentage of bare ground on the campsite | 2 (4%) |
| Level of water visibility | 2 (4%) |
| Percentage of vegetation coverage | 2 (4%) |
| Amount of human impact at camping sites | 1 (2%) |
| Percentage of area erosion | 1 (2%) |
| Presence/extent of trails | 1 (2%) |
| Family biotic index less of than five | 1 (2%) |
| **Managerial** | |
| Entrance fee charge | 2 (4%) |
| Distance of settlement from the river | 1 (2%) |
| Number of camping bays | 1 (2%) |
| Number of houses along the shore | 1 (2%) |
| Number of parking bays | 1 (2%) |
| Number of signs | 1 (2%) |

**Table 7.** *Cont.*

| Carrying Capacity Assessment | Frequency (%) |
|---|---|
| **Carrying capacity as an index or value** | |
| Carrying capacity index | 1 (2%) |
| Tourism economy carrying capacity value | 1 (2%) |
| Ecological carrying capacity value | 1 (2%) |
| Resource carrying capacity index | 1 (2%) |

Note. n = 50.

## 4. Discussion

This systematic quantitative literature review evaluated 100 original research papers from three repositories of scientific publications. The analysis provides insight into the methods used to assess carrying capacity, trends, locations, settings, and assessments of carrying capacity that were made in the past two decades. Four main findings from the analysis were identified: the differences between the methods of assessing carrying capacity, limitations, issues, and the gaps in carrying capacity research.

### 4.1. The Differences between the Methods of Assessing Carrying Capacity

The normative approach and Cifuentes Arias' method were the two most commonly used methods in assessing carrying capacity. As shown by the publication trend, Cifuentes Arias' method was slightly "younger" or "more active" compared to the normative approach, since 90% of publications using this method were published post-2011. Cifuentes Arias' method was pioneered by Cifuentes Arias [59,60], and the normative approach originated from visitor management frameworks developed in the USA, such as Limit Acceptable Change (LAC), Visitor Experience Resource Protection (VERP), and Visitor Use Management (VUM). The normative approach was predominantly utilized in the USA and in wilderness settings, whereas Cifuentes Arias' method was employed mainly by studies in Brazil, China, and Iran in beach and urban environments.

A total of 24 methods of assessing carrying capacity were developed based on different procedures. Each method differed from the others, and there was no standardized formula for determining the carrying capacity [61]. However, these methods shared a similar goal based on two different fundamentals of carrying capacity. The first fundamental put weight on the maximum number of visitors, and the second accentuated the level of acceptable use.

Cifuentes Arias' method consists of three sequential formulas: PCC, RCC, and ECC. The formulation of PCC considers the size of the recreation setting and the area required for one visitor per $m^2$ (V/a) to find the acceptable number of users that can be accommodated in an area of a recreational setting. The use of V/a varies depending on the activity type and area size [62–65].

The second formula is the RCC, sometimes used and known as Ecological Carrying Capacity. The formula is the maximum permissible number of visits to a site once the corrective factors have been applied to the PCC. Corrective factors can be in the form of environmental conditions such as the rainy and windy season [62,64–69], ecological conditions that are vulnerable to human use [62,64,66,67,70], and wildlife mating season [66]. By applying the corrective factors, the number of visitors that can physically accommodate an area (PCC) is further optimized based on RCC.

The last formula of Cifuentes Arias' method is the ECC, also known as Tourism Carrying Capacity (TCC). The number of people or visitors determined via RCC is again optimized by considering management capacities, such as the number of personnel available [64–66,69,71], infrastructure and equipment [67,70], and capacity index [63]. The product of Cifuentes Arias' method is almost solely based on the maximum number of visitors (per day/month/year/at one time). Therefore, it is associated with a definition of carrying capacity that emphasizes the maximum/optimum/acceptable number of tourists

or visitors that may visit a destination without causing damage to the physical environment, as well as visitor experience satisfaction [72].

Other methods that have a similar foundation of carrying capacity are a proportion curve and regression model [73,74], the optimum boating capacity formula [75,76], a fuzzy coefficient and linear programming model [77,78], and the tourism environmental carrying capacity formula [79]. These methods tap the perspective of residents' attitudes and quality of life [80,81] in determining the carrying capacity and thus could also be grouped as methods that gravitate toward the maximum number of people acceptable in tourist destinations.

In contrast, the normative approach method is based on the second fundamental of carrying capacity that emphasizes how much recreational use or the amount of use that can be allowed to address the carrying capacity [61,82]. Originating from visitor management frameworks, carrying capacity based on this method emphasizes at least three considerations before it can be determined. The first is that carrying capacity is a product of value judgment [61,83–90]. Second, it depends on management objectives, desired conditions, budget allocation, and available staff [61]. The third is that the carrying capacity is determined by management zones' objectives, and the carrying capacity of each zone may vary depending on the objectives specified for the zone. Lastly, it requires the determination of indicators or indicators of quality and the specification of a standard or a standard of quality.

The key takeaway of this method is that it requires identifying indicators and specifying the standards that define the quality of the recreation resource [34,36–48,61,85,91,92]. The indicators could be determined by examining the state of the recreation and tourism resource quality, such as the environmental and social impacts as well as the managerial influences that are likely to be the factors that would influence visitor experiences [93–95]. A standard is principally determined by norms, i.e., typical, common, or generally accepted references within a social context [14,16,91,96–107].

Norms play an essential role in finding a consensus among users, stakeholders, and management on a standard for indicators. Hence, it is the product of a value judgment. A visual approach, such as computer-edited photographs showing a range of conditions related to the indicators, is often utilized to assess the norms, i.e., the standards. The norms are then plotted using the return potential model (social norm curve) [108] to determine the amount of use acceptable for the carrying capacity indicators.

In addition, indicators and standards play essential roles in monitoring purposes [85,86,91]. The indicator variables can be tracked over time to ensure that the carrying capacity is within the standard [91]. Other methods with a history of being employed along with the normative approach are methods using crowding [109–113] and indifference curve analysis to determine trade-offs between the amount of recreational use and the surroundings [114]. In addition, methods such as the tourism carrying capacity index [115], choice experiment methods [116,117], the tourism carrying capacity formula [118], the time-for-time substitution model, and individual-based and system dynamics models [119] are also aligned with the second fundamental of carrying capacity, as these methods are developed to monitor changes related to tourism destinations' environmental, social, and managerial conditions.

## 4.2. Limitations of the Two Fundamentals of Carrying Capacity

Based on the documented methods, it is apparent that there are two different fundamentals of carrying capacity. The differences lie in the tenets of carrying capacity that are applied. First, carrying capacity is seen as the capacity of a tourism destination or a recreation resource to hold/tolerate/absorb visitors or activities before changes to the physical and social environment and visitor experience occur [117]. Given the output of the measurement and formula used, a carrying capacity that is based on this fundamental postulate shows that the level of use directly influences the degradation of the physical environment and visitors' experiences.

Nevertheless, carrying capacity based on such fundamentals has been criticized for depending on a "magic number" to solve the issues surrounding tourism and recreation resources [88,120]. The main criticism is that the link between the amount of use and the impact on the physical environment is not always direct, as degradation of the physical environment or recreation or tourism impacts are also influenced by the timing of use, type of use, use distribution, setting types, and management actions [85,86,120,121]. Advances in the recreation ecology field of research have demonstrated that the magnitude of the recreational impact is more prevalent during its initial development [122,123]. Ergo, a carrying capacity aligned with limiting the amount of use is considered ineffective [85,86,91,124,125], as it could not generate stability between resource protection and visitor use [86,126–128].

Thus, carrying capacity based on the amount of acceptable use has been developed, since changes in environmental conditions or visitor experience will occur nonetheless [85,91], even at a low or high level of use. Sooner or later, the changes will become unacceptable. This foundation of carrying capacity puts weight on indicators and standards by defining the level of environmental protection to be maintained and the type of visitor experience to be offered. It is also important to note that via visitor management frameworks such as the LAC and VERP, the involvement of stakeholders and visitors is considered in formulating the level of use that would be acceptable for the recreation and tourism settings.

The method associated with the amount of use has, however, been criticized for the issue of displacement. Visitors who are daunted by increasing use levels in an area tend to move to other areas, and they are replaced in the former area by users who have a greater tolerance for environmental, social, and managerial changes [54]. However, visitor displacement could be managed via management strategies or used as the indicator's variables [129–131]. In addition, norms associated with the level of use in recreation settings have been found to be stable over time [99,132,133].

### 4.3. Issues in the Carrying Capacity of Tourism Destinations

Carrying capacity was introduced in recreation resources in the 1960s and was soon adopted into the tourism context [54]. In the earlier phase of its development, carrying capacity was defined as the maximum number of people an area can sustain without impairing the integrity of its resources. The definition was then extended to include an experiential component, as environmental impacts also threatened visitors' experiences. The current review revealed that the methods used to formulate carrying capacity in tourism destinations and recreation resources are interchangeable.

Nonetheless, some methods were developed specifically for tourism destinations (e.g., [77,78,115,116,119]). Carrying capacity based on these methods was formulated at the regional level involving enormous data encompassing economic indicators (e.g., tourism income, labor force, and per capita GDP), infrastructure (e.g., accommodations, restaurants, and road networks), ecology and resources (e.g., water quality, air quality, and solid waste), and social indicators (e.g., tourist satisfaction and resident satisfaction). The tourism carrying capacity developed based on such indicators may be beyond the control of management or authorities, as it requires political will and policy changes to ensure its effectiveness.

On the other hand, carrying capacity from the recreation perspective revolves around resources (e.g., campsite impacts, trail impacts, and litter), experiential issues (e.g., crowding), and managerial conditions (e.g., permits offered, facilities provided) of the location under investigation. Carrying capacity using recreation resources was developed mainly for the local conditions. Thus, managing such a capacity could be within the management's capability.

### 4.4. Gaps in Carrying Capacity Research

This review has revealed that many carrying capacity studies have been carried out in the established area. Whittaker et al. [61] have argued that carrying capacity can be

formulated proactively before impacts occur. Thus, there is a shortcoming in the literature that warrants the need to develop a comprehensive method for formulating the carrying capacity for a newly established recreation and tourism destination area. Moreover, the methods of assessing carrying capacity that have been used over the past two decades have centered on formulating the capacity, thus leaving a gap in the literature regarding the research and procedures for carrying capacity monitoring, which is crucial for the sustainability of a recreation and tourism destination.

In addition, this review has discovered that the most attention has been given to the social aspects of carrying capacity, which account for nearly 70% of the total carrying capacity assessments recorded in this study. The carrying capacity is governed by the resource, experiential (social), and managerial conditions of recreation resources [46]. Similarly, the nature-based recreation experience model [134] states that social elements, components of resources (i.e., nature-oriented details and scenic values), and management influences shape the quality of visitor experience. Further, the carrying capacity, as is its universal definition, emphasizes the importance of the physical environment and the quality of tourist satisfaction (see [135]). Thus, resource and managerial conditions are equally important in a recreation and tourism destination in addition to the social aspects.

The type of resource assessment most frequently recorded in this study focuses on campsites (e.g., the bare ground on a campsite, the area erosion, and the number of human impacts) and on the threshold or standards for litter accumulation. Only a few studies were designed to formulate the carrying capacity of resource conditions, especially on trails. Given the growth of ecotourism destinations in many countries, the length and spread of trail networks could be expanding in many of the recreation and tourism resources worldwide [58]. Additionally, a staggering 30 million users from 190 countries registered as users of the AllTrails mobile app in 2021 [136]. This enormous number of users warrants more research regarding the carrying capacity of the trail's resources and managerial conditions for more sustainable use of recreation and tourism settings.

Recreational impacts on soil, vegetation, and wildlife have been well researched, especially in the USA and Australia. However, the magnitude and types of impacts can be ecosystem-specific [18,137]. In addition, most of the research on recreational impacts looks at urban and temperate broadleaf and mixed forest, and only 1% has focused on tropical area [58]. Similarly, this study has revealed very few carrying capacity studies in tropical regions; these are limited to Indonesia, Malaysia, Sri Lanka, and Puerto Rico. Since the formulation of carrying capacity goes in tandem with recreational impact assessment, more resource-oriented carrying capacity studies need to be carried out in tropical regions.

Another area of focus for carrying capacity studies is that of threatened plant communities [58]. As suggested, such areas are underrepresented in the existing impact studies, especially in visitor-laden areas such as mountaintops, waterfalls, cliffs, or areas in which rare species of vegetation are found. Furthermore, the standard of the night sky is another resource-based carrying capacity that needs attention. The issue of the disappearing night sky due to light pollution, particularly in urban and rural environments, has encouraged more people to explore wilderness settings to enjoy the night sky. Light pollution from surrounding areas and excessive outdoor lighting in these settings could jeopardize the opportunity to view the night sky in the wilderness. Although Manning et al. [138] provided one of the earlier efforts to formulate a standard of the night sky in Acadia National Park, USA, more research on the night sky standard is undoubtedly needed in many recreation and tourism resources worldwide.

## 5. Conclusions

### 5.1. Study Limitations

First, this study was confined by the limited number of repositories in the search process. Repositories such as the Web of Science and Academic Search Complete are the two most common repositories used for systematic literature review research [49,50,56,139]. However, this study was limited to the repositories to which the institution in which

this study was carried out subscribes. Nonetheless, this study was able to document 100 original research papers despite such a limitation, which is comparable to other systematic literature review studies [56,58].

Second, this study was limited by the number of keywords used for the search. Keywords that were excluded from this review, such as psychological/perceptual carrying capacity, economic carrying capacity, carrying capacity index, sustainable recreation, and tourism management, as well as other types of publications that were also excluded, could uncover other dimensions of carrying capacity studies that were not discussed in this review.

Third, this study was constrained to a review of the methods used for carrying capacity determination. It would be interesting for future systematic literature review studies to record and discuss the variation in the definitions of carrying capacity as well as in the carrying capacity types that encapsulate carrying capacity research. For example, social carrying capacity sometimes refers to the maximum level of recreational use beyond which the quality of the recreational experience depreciates from the perspective of visitors [140,141] or the host population's tolerance level for tourists' presence and behavior in the destination area [142,143].

Fourth, this study was bound to publications that were published in the English language. Non-English language research papers were excluded, as translation into the English language was beyond the authors' expertise. In addition, excluding non-English language research papers may not alter the trend of scientific studies [58] or result in a bias [144]. The results of this study have shown that carrying capacity studies have been published by countries with different official languages. Therefore, it may be assumed that the research papers reviewed in this study cover most of the significant publications on carrying capacity studies.

### 5.2. Contributions, Implications, and Future Studies

This review has yielded several significant findings for future carrying capacity studies. The first is associated with the carrying capacity fundamental. Carrying capacity related to limiting the maximum number of people in an area has been criticized by many researchers in the past [86,91,120,128]. This criticism has led to the development of carrying capacity fundamentals that accentuate the amount of acceptable use, as outlined in several management frameworks such as the LAC, VERP, and VUM. Despite this criticism, this fundamental carrying capacity is receiving more attention lately, as nearly 60% of the studies reviewed were published in the last five years. Since the fundamental of a carrying capacity is encapsulated by its definition, this study calls for improving the definition of carrying capacity by emphasizing the importance of the amount of acceptable use and monitoring environmental and experiential changes via indicators and standards of carrying capacity.

The second finding of this study is the need to develop an approach to determine carrying capacity ahead of environmental and experiential changes. As shown in the results, the carrying capacity studies that were reviewed were conducted in the already established recreation and tourism resources area. The available methods depend on the presence of visitors and the degree of impact on the site, as the inventory and assessment of environmental changes are prerequisites in specifying carrying capacity. However, in a newly established area in which the presence of visitors and impacts are absent, or for the case in which the base data regarding the impacts or the changes to the environmental conditions are unavailable, the formulation of carrying capacity needs to be addressed. It has been argued that formulating carrying capacity ahead of time requires thoughtful consideration and professional judgment [61]. Alternatively, a new method could be developed, or the available methods could be integrated to specify the capacity ahead of time for newly established areas of tourism and recreation resources. For example, Cifuentes Arias' method and the normative approach could be integrated to determine capacity related to crowding. Nonetheless, the formula must be adjusted to accurately reflect the onsite condition.

The third finding is the need for more research on carrying capacity that reflects the resources and managerial conditions of recreation and tourism resources. Most of the reviewed studies center on social conditions. Thus, more research on carrying capacity with respect to resource and managerial conditions is needed, as both are vital components of recreation and tourism settings. Granted, some of the methods employed in the papers here gravitate toward the maximum number of people in an area by considering resource, social, and economic limitations. However, since changes are inevitable, indicators and standards/thresholds of the resource and managerial conditions should be established to ensure that these conditions can be monitored over time.

Concerning managerial implications, as shown in the review, establishing carrying capacity requires comprehensive base data (e.g., resource conditions, visitation growth, level of visitor experience quality, and management capacities). Thus, developing strategies for time-series base data development, especially on the resource's conditions, is crucial to establish a sound carrying capacity that reflects the conditions onsite. Furthermore, as emphasized by the carrying capacity based on the acceptable use amount, carrying capacity should be employed with indicators for monitoring purposes. Monitoring indicator variables (e.g., resource, experiential, and managerial conditions) is imperative to ensure that the standards/thresholds of carrying capacity are acceptable.

Fourth, more research is needed or methods could be developed to monitor the effectiveness of the carrying capacity that has been implemented. In addition, carrying capacity in a tourism destination is formulated based on comprehensive data such as the economic, infrastructure, ecological/resource, and social indicators. As these indicators involve different parties with different aims and needs, implementing such a capacity may require political and policy changes. Ergo, future studies that monitor the indicators are crucial for protecting natural resources, maintaining sustainable tourism, balancing economic benefits and environmental impacts, addressing overcrowding, and supporting local communities.

*5.3. Conclusions*

This study reviewed 100 original peer-reviewed carrying capacity research papers published from 2000 to October 2022. Twenty-four methods were recorded, with the normative approach and Cifuentes Arias' method being the two primary methods used in determining the carrying capacity of recreation and tourism resources. Carrying capacity studies have been carried out in previously established areas with issues related to environmental and experiential changes present onsite. This systematic literature review has also showed that the methods used to assess carrying capacity were governed by two fundamentals. Carrying capacity associated with a maximum/acceptable/optimum number of visitors/tourists in an area has been criticized for depending on a "magic number", whereas the normative approach has been criticized for the issue of displacement. However, the two primary methods of assessing carrying capacity could be combined, especially in a newly established area in which the carrying capacity could be implemented before recreational impacts take place, especially regarding issues related to crowding. In addition, carrying capacity in tourism is based on complex data, where some of the data or indicators are outside the control of the destination and thus may require political will and policy changes to be implemented. Therefore, monitoring the effectiveness of the carrying capacity implemented is crucial to ensure its effectiveness. Most of the carrying capacity assessments noted have focused on social aspects. As resource and managerial conditions are also components of carrying capacity, monitoring the changes in these two conditions is vital for the sustainability of the recreation and tourism destination. As such, more studies are needed regarding the resource and managerial aspects of carrying capacity.

**Supplementary Materials:** The following supporting information can be downloaded at: https://www.mdpi.com/article/10.3390/su15043474/s1.

**Author Contributions:** Conceptualization by Z.A., A.A., S.S.N.Y., M.M. and S.A.B.; literature review, data analysis, writing—original draft preparation, Z.A. and S.A.B.; writing—review and editing, Z.A., A.A., S.S.N.Y., M.M. and S.A.B. All authors have read and agreed to the published version of the manuscript.

**Funding:** This research received no external funding.

**Institutional Review Board Statement:** Not applicable.

**Informed Consent Statement:** Not applicable.

**Data Availability Statement:** Not applicable.

**Conflicts of Interest:** The authors declare no conflict of interest.

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
