# Peer review of "Systematic Literature Review on Methods of Assessing Carrying Capacity in Recreation and Tourism Destinations"

_sustainability, doi:10.3390/su15043474_

Round 1

Reviewer 1 Report

The paper is good , although the limits as the authors wrote in the papers.

Author Response

Thank you for allowing us to submit a revised version of our manuscript. We appreciate the time and effort in providing feedback on our manuscript and are grateful for the insightful comments in improving our paper. 

Reviewer 2 Report

The idea of this review looked promising; unfortunately, the study did not live up to my expectations. Even after reading the study to the end, I have no idea what the authors attempted to review. I believe the main problem with this study is that it lacks a clear structure and organization. Also, many sentences lack logic. For example (lines 38-39):

"Recreation and tourism resources are universally protected natural settings preserved for protecting some of the earth's wonders and natural history". 

Other issues

Line 54: Why do you start the sentence with "Thus"?

Lines 143-144: "Since 2000, publications have peaked twice between 2012 and 2011 (f= 8 each) and 2021 (f=9)". What do you mean?

Lines 150-151: "Six first authors contributed two research papers". How is this possible? If there are six first authors there should be at least six papers (same at lines 193-195). Or do you mean, two research papers each?

Lines 178-179: You mention 24 methods of carrying capacity. Do you mean methods of assessing (measuring) carrying capacity?

Discussion section: At the beginning, it regurgitates information from the analysis of the results section.

Appendix A: it says: "100 included and 145 excluded sources". I do not know what this is supposed to mean especially since the appendix lists 144 sources.

You did not explain why you decided to start your review with year 2000. Did you find previous studies that ended their review in 1999 and you wanted to continue from where they left? Did you want to review the literature on carrying capacity published over the last 23 years? Plus, 2022 has not ended.

However, most of these are minor issues. The main problem is that we are not told what this study is supposed to review: how the understanding of the concept has evolved in time, methods of assessing carrying capacity? Even the title is very ambiguous.

Author Response

Thank you for allowing us to submit a revised version of our manuscript. We appreciate the reviewers' time and effort in providing feedback on our manuscript and are grateful for the insightful comments in improving our paper.

We have incorporated most of the suggestions made by the reviewers, and those changes are highlighted in the revised manuscript. Please see below for a point-by-point response to the reviewers' comments. All page numbers refer to the revised manuscript file with edited text in red font.

We thank the reviewers again for taking the time to review our manuscript. We look forward to hearing from you regarding our submission and responding to any further questions and comments you may have.

Reviewer 3 Report

See attached file

Author Response

Thank you for allowing us to submit a revised version of the manuscript. We appreciate the reviewers' time and effort in providing feedback on the manuscript and are grateful for the insightful comments in improving the paper.

We have incorporated most of the suggestions made by the reviewers, and those changes are highlighted in the revised manuscript. Please see the attached document and the revised manuscript for a point-by-point response to the reviewers' comments. All page numbers refer to the revised manuscript file with edited text in red font.

We thank the reviewers again for taking the time to review our manuscript. We look forward to hearing from you regarding our submission and responding to any further questions and comments you may have.

Reviewer 4 Report

The theme of the article is very important, but - apart from literature review - some outstanding scientific contribution is missing.

Author Response

Thank you for allowing us to submit a revised version of the manuscript. We appreciate the editor's and reviewers' time and effort in providing feedback on our manuscript and are grateful for the insightful comments in improving our paper.

We have incorporated most of the suggestions made by the reviewers, and those changes are highlighted in the revised manuscript. Please see below for a point-by-point response to the reviewers' comments. All page numbers refer to the revised manuscript file with edited text in red font.

We thank the reviewers again for taking the time to review our manuscript. We look forward to hearing from you regarding our submission and responding to any further questions and comments you may have.

Reviewer 5 Report

1. In the abstract, it is suggested to add some background with few objectives and possible applications of this study and highlight the novelty of this work clearly. 2. Problem formulation is not adequate. The authors must substantiate the contributions of the paper vis-a-vis the gaps in the literature. Authors can include the following https://onlinelibrary.wiley.com/doi/abs/10.1111/exsy.12232 3. The inclusion and exclusion criteria as per PRISMA framework is Ok. However, the in the diagram, it is mentioned that total number of papers screened = 100; exclusion = 60. Then how can there be 100 papers in the sample to study? 4.  Future direction section is weak. Please elaborate. 5. Add to managerial and/or social implications of the study.  

Author Response

Thank you for allowing us to submit a revised version of the manuscript. We appreciate the reviewers' time and effort in providing feedback on our manuscript and are grateful for the insightful comments in improving our paper.

We have incorporated most of the suggestions made by the reviewers, and those changes are highlighted in the revised manuscript. Please see below for a point-by-point response to the reviewers' comments. All page numbers refer to the revised manuscript file with edited text in red font.

We thank the reviewers again for taking the time to review our manuscript. We look forward to hearing from you regarding our submission and responding to any further questions and comments you may have.

Round 2

Reviewer 2 Report

Looks much better now. My main concerns were answered.

Best of luck.

Reviewer 5 Report

1. Provide the highlights more clearly.

2. The linking of the overall article appears discretely, not providing the idea of the paper, even within section. 

3. Improve the quality of English writing.

Round 3

Reviewer 5 Report

Most of the previous comments are addressed by the authors. Future research direction could have been done more appropriately. Check the grammatical errors carefully.  
